# Dislodgement Forces and Cost Effectiveness of Dressings and Securement for Peripheral Intravenous Catheters: A Randomized Controlled Trial

**DOI:** 10.3390/jcm9103192

**Published:** 2020-10-01

**Authors:** Axel Schmutz, Lea Menz, Stefan Schumann, Sebastian Heinrich

**Affiliations:** Department of Anesthesiology and Critical Care, Medical Center, Faculty of Medicine, University of Freiburg, 79106 Freiburg, Germany; Lea.Menz@t-online.de (L.M.); stefan.schumann@uniklinik-freiburg.de (S.S.); sebastian.heinrich@uniklinik-freiburg.de (S.H.)

**Keywords:** peripheral intravenous catheterization, dressing, securement, vascular access devices, catheter dislodgement

## Abstract

Objectives: Peripheral intravenous catheters (PIVC) are the most frequently used invasive devices in medicine. PIVC failure before treatment completion is a significant concern and occurs in 33–69% of patients. Partial dislodgement and accidental removal are some of the reasons for PIVC failure. The most effective dressing and securement method for preventing accidental removal remains unclear. It was the aim of this study to compare the force required to dislodge a PIVC with four commonly used dressing and securement methods. Additionally, costs were calculated. Methods: Truncated 18-gauge i.v. cannulas were attached onto the forearm of 209 volunteers using four different dressings and securements (sterile absorbent wound dressing covered by two different types of elastic polyester fleece, bordered and non-bordered polyurethane). The force during continuously stronger pulling until dislodgement was recorded. Results: The highest resistance against dislodgement forces could be observed with a sterile absorbent wound dressing covered by two incised elastic polyester fleece dressings. Commercially-manufactured bordered and non-bordered polyurethan film dressings were 20% to 75% more expensive than sterile absorbent wound dressings covered by elastic polyester fleece dressing. Conclusions: Elastic polyester fleece secured a PIVC against accidental removal by external force best, compared to commercially-manufactured bordered and non-bordered polyurethane film dressing.

## 1. Introduction

Peripheral intravenous catheters (PIVC) are the most frequently used invasive medical devices and are routinely inserted during emergency treatment, anesthesia and critical care. They are an essential component for the delivery of medication and fluid resuscitation. Moreover, almost all hospitalized patients have a PIVC at some point. Worldwide annual usage of PIVC exceeds two billion units [1] and up to 330 million PIVCs are sold annually in the United States [2].

PIVC failure before treatment completion is a significant concern and its frequency ranges from 33% to 69% [3]. Besides phlebitis, occlusion and infection, partial dislodgement and accidental removal are some of the reasons for PIVC failure. Dislodgement is reported to occur in 7–10% of adult patients from medical-surgical non-ICU departments [1,4] and in up to 10–15% of patients in emergency departments [4]. PIVC dislodgement and replacement puts patients at risk of hematoma, delays intravenous therapy, drives costs for the institution and may cause pain and anxiety in patients, especially in those with difficult venous access. Dressing and securement of a PIVC are supposed to reduce movement of the catheter at the insertion site, and reduce the risk of external contamination. There are guidelines that address the fixation method with regard to the prevention of catheter-associated infections. However, the issue of the optimal PIVC dressing and securement method is still considered unresolved [1]. Given the large number of PIVCs used in hospitals, the strain of financial resources is an additional concern.

There is a paucity of data reporting the efficacy of securement methods for PIVC in regard to dislodgement force. Found and Baines tested different taping methods to secure a cannula to a polyvinylchloride pipe and to one single volunteer [5]. However, in this study, no sterile dressing was attached although it is recommended in clinical practice guidelines [6,7]. In a recent Cochrane systematic review, transparent dressings, bordered and non-bordered, were more effective in preventing dislodgement or accidental removal than a fixation with gauze or a securement device [3].

Nevertheless, the most effective dressing and securement method for preventing accidental removal remains unclear.

It was the aim of this study to determine the maximum force required to dislodge a PIVC with four commonly used dressing and securement methods. Additionally, the respective costs of the four investigated dressing and securement methods were calculated.

## 2. Materials and Methods

### 2.1. Study Institution and Ethics

The study was conducted at the Department of Anesthesiology and Critical Care, Medical Center, University of Freiburg, Freiburg, Germany, after approval of the local Ethics Committee (August 28 2019, Approval Number EK 289/19). The study was registered in the German Clinical Trials Register (DRKS00018106).

### 2.2. Subjects

After ethical approval and obtaining written informed consent, a total of 209 volunteers were included in the study. Exclusion criteria were non-intact skin on forearm or antecubital region, steroid medication and age < 18 years.

### 2.3. Cannula, Securements and Dressings

The intravascular part of an 18-gauge i.v. cannula with an injection port (Vasofix^®^ Safety, B Braun, Melsungen, Germany) was truncated (Figure 1) and the remaining part fixated at the forearm via one of the four tested securements and dressings. All securements and dressings were attached by the same investigator (L.M.).

The following four securements and dressings were evaluated (Figure 1):(a)A sterile absorbent wound dressing (Leukomed^®^, BSN medical, Hamburg, Germany; 7.2 × 5 cm) placed onto the insertion site, covered and overlapped by two rectangular elastic polyester fleece dressings (Fixomull^®^ stretch, BSN medical, Hamburg, Germany). One of these dressings was incised to secure the wings of the i.v. cannula. The second dressing was taped without incision.(b)A sterile absorbent wound dressing (Leukomed^®^, BSN medical, Hamburg, Germany; 7.2 × 5 cm) placed onto the insertion site, covered and overlapped by two rectangular elastic polyester fleece dressings (Fixomull^®^ stretch, BSN medical, Hamburg, Germany). Both of these dressings were incised to secure the wings of the i.v. cannula.(c)Transparent polyurethane film dressing (3M™ Tegaderm™ I.V., 3M, St.Paul, MN, USA; 7 × 8.5 cm).(d)Nonwoven polyester with polyurethane film laminate (Curafix^®^ i.v. control, L&R, Frankfurt, GERMANY; 9 × 6 cm).

The skin areas covered by the securements were 60 cm^2^ (a and b), 59.5 cm^2^ (c) and 54 cm^2^ (d) and were therefore comparable.

### 2.4. Force Measurement

A Luer lock adapter was attached to the Luer lock plug of the truncated 18-gauge i.v. catheter. The free end of the Luer lock adapter was attached to a cord that was arranged at an angle of 30° to the horizontal. The cord was then passed through two deflection rollers and attached to a force transducer (PSD-S1 Load cell, Zhengzhou Pushton Electronic Instruments Co., Ltd., Zhengzhou, China). Continuously increasing tension on the catheter securement was applied by turning a crank handle attached to the force transducer (Figure 2). The peak force occurring before dislodgement was determined from continuous force recordings.

### 2.5. Procedure

After preparation and disinfection of hairless forearm skin (Octeniderm^®^, 0.1% octenidin in 75% isopropyl alcohol, Schülke, Norderstedt, Germany), every securement method was applied sequentially onto the above described i.v. catheter on each subject. The sequence of securement methods was randomized by a list randomizer (www.random.org) ahead of the study. After attaching the randomized securement method, the pulling force on the securement was continuously increased by turning the crank handle until the securement dislodged.

### 2.6. Retest Reliability

To determine the level of precision of the force sensor, the dislodgement forces of the dressings were measured repetitively under identical conditions (same subject) in a preliminary experiment. Each dressing was allocated to one author’s forearm as described above and was dislodged 20 times in a row from the identical position (antecubital region). The retest-reliability of all tested securements was comparable and did not show systematic trends (Figure 3).

### 2.7. Endpoints

The primary endpoint of the study was the maximum dislodgement force. Costs were calculated for each securement method. Cost calculations were based on 2020 prices in euros. Costs for staff time to apply securements were not considered.

### 2.8. Data Processing and Statistical Analysis

Raw data were collected and analyzed via software self-written in LabVIEW 7.1 (National Instruments, Austin, TX) and MATLAB (MathWorks^®^, Natick, MA, USA) and transferred to R (R Foundation for Statistical Computing, Vienna, Austria) for statistical processing. The minimum number of subjects was determined by a sample size calculation carried out ahead of the study. The number of cases to achieve a desired power of 95% at an alpha error of 5% was determined to be 208. The bases for the sample size calculation were preliminary test runs with the authors as subjects for the evaluation of the test set-up. Continuous variables were examined for normal distribution using the Shapiro–Wilk test. Subsequently, one-way analysis of variance was used, followed by Fisher’s Protected Least Significant Difference (PLSD) testing post hoc test, if the ANOVA indicated significant differences between means.

## 3. Results

A total of 209 subjects gave written informed consent to participate in the study. The subjects’ characteristics are shown in Table 1.

### 3.1. Force Measurements

Force measurements showed the force characteristically increasing with time until the dressings detached from the skin (Figure 4).

### 3.2. Dislodgement Forces

The distribution of dislodgement forces for the different dressings showed Gaussian distribution profiles for both polyester fleece dressings and the transparent polyurethane film dressing (Figure 5).

The highest dislodgement force could be observed with the sterile absorbent wound dressing covered by elastic polyester fleece dressings which was about twice as high as the lowest dislodgement force determined for polyester with polyurethane film laminate (Figure 6).

### 3.3. Age and Gender Differences

Age was a significant factor for dislodgement force (*p* = 0.0215), but the correlation was very weak (r^2^ = 0.006) and therefore clinically not relevant. There were no significant differences in dislodgement forces between women and men.

### 3.4. Costs

Commercially-manufactured bordered (Curafix^®^ i.v. control) or non-bordered (Tegaderm™ I.V.) polyurethan film dressings were 20% and 75% more expensive, respectively, than sterile absorbent wound dressings covered by elastic polyester fleece dressing (Table 2).

## 4. Discussion

The most important finding of our study is that a sterile absorbent wound dressing covered by two elastic polyester fleece dressings is superior in terms of both fixation security and material costs compared to commercially-manufactured bordered and non-bordered polyurethane film dressings.

During emergency and critical care and during anesthesia, it is of utmost importance to have a secure i.v. catheter. Patient movement, positional changes, transfer from the stretcher onto the CT scanner or into the ICU bed may generate pulling forces on the i.v. lines and, finally, dislodge and inadvertently remove the PIVC.

In a recent international cross-sectional study of more than 40,000 PIVCs in 51 countries, the primary dressing material used (78%) was transparent film dressing [8]. Visibility of the insertion site is an essential advantage of transparent film dressing and may reduce dressing changes. Transparent film dressings were also described to be more effective in preventing dislodgement or accidental removal compared with gauze or a securement device in a systematic review [3]. In contrast to this finding, our results show significantly lower dislodgement forces with bordered and non-bordered transparent film dressings compared to sterile wound dressings and elastic polyester fleeces. A further increase in dislodgement forces may be generated by increasing the skin area covered by elastic polyester fleece, a measure not possible with commercially available dressings. Further trials are needed to evaluate a combination of transparent film dressing and elastic polyester fleece.

A randomized controlled trial using skin glue to secure PIVCs in the emergency departmental setting showed a 10% reduction in PIVC failure rate and the authors speculate that this effect may have been caused by a lower rate of dislodgement [9]. However, the costs of skin glue greatly exceed the expenditure for dressings and securements [10].

Older patients often have poorer skin elasticity. Thereby, they may suffer skin lacerations by stretching and pulling forces. Further trials are needed to study dressings and securements in this population.

### Limitations

This single-center randomized trial design was inevitably not blinded. The dressings and securements were attached only for a short period of time until the pulling force was applied. Therefore, the effects of soiling, sweating and moisture were not taken into account. We only examined longitudinal traction forces at an angle of 30° to the horizontal, although the various potential directions of forces make it difficult to study this issue further. We did not place the i.v. catheter into the veins of the subjects. However, we would not expect that an additional intravascular part of the catheter would have changed our results by adding a relevant resistance to the pulling forces and further—this condition was the same in all experiments. We did not calculate costs including staff costs. Moreover, only one person applied the dressings, presumably not representative for the average clinical user. However, staff costs may not be generalizable to other country settings and were therefore not taken into account.

## 5. Conclusions

Our study shows the existence of a favorable combination of sterile gauze and elastic polyester fleece to secure a PIVC against accidental removal. Decreasing the failure rate by this simple dressing method can improve patient care and safety and material costs are low. We therefore suggest applying this dressing and securement combination.

## Figures and Tables

**Figure 1 jcm-09-03192-f001:**
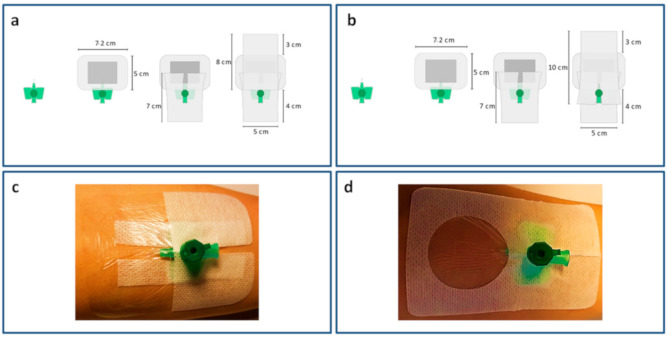
Four securements: (**a**) sterile wound dressing covered by two elastic polyester fleece dressings, one incised; (**b**) sterile wound dressing covered by two elastic polyester fleece dressings, both incised; (**c**) non-bordered transparent polyurethane film dressing; (**d**) bordered polyurethane film dressing (nonwoven polyester with polyurethane film laminate).

**Figure 2 jcm-09-03192-f002:**
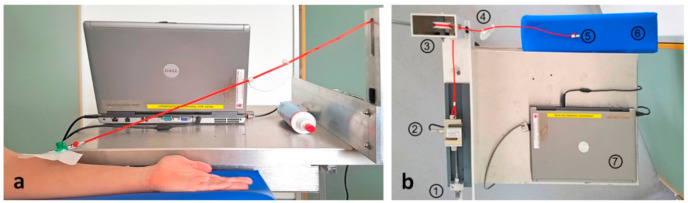
Experimental setup: (**a**) cord attached to truncated cannula; (**b**) view from above: 1, crank handle; 2, force transducer; 3, deflection roller; 4, angle meter; 5, Luer lock connector; 6, arm rest; 7, laptop for data processing.

**Figure 3 jcm-09-03192-f003:**
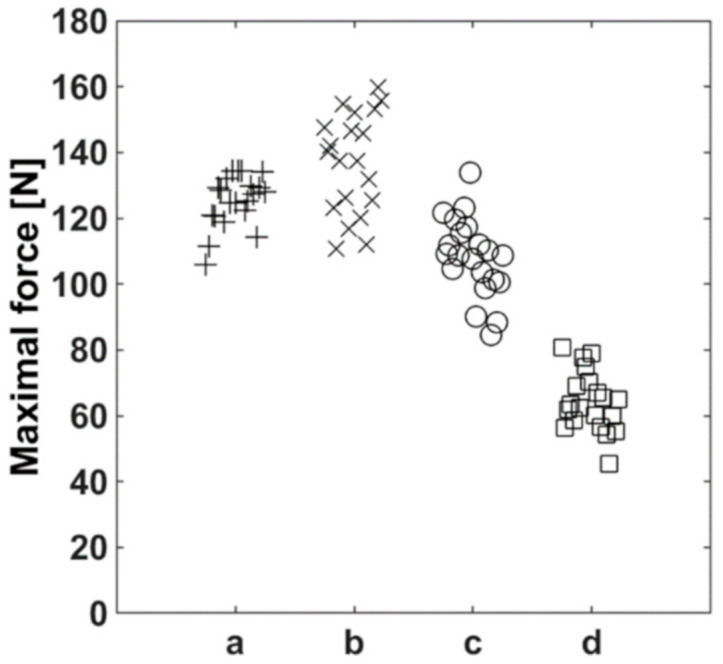
Retest-reliability of the four securements (20 measurement repetitions). (**a**) Sterile wound dressing covered by two elastic polyester fleece dressings, one incised; (**b**) sterile wound dressing covered by two elastic polyester fleece dressings, both incised; (**c**) non-bordered transparent polyurethane film dressing; (**d**) bordered polyurethane film dressing (nonwoven polyester with polyurethane film laminate).

**Figure 4 jcm-09-03192-f004:**
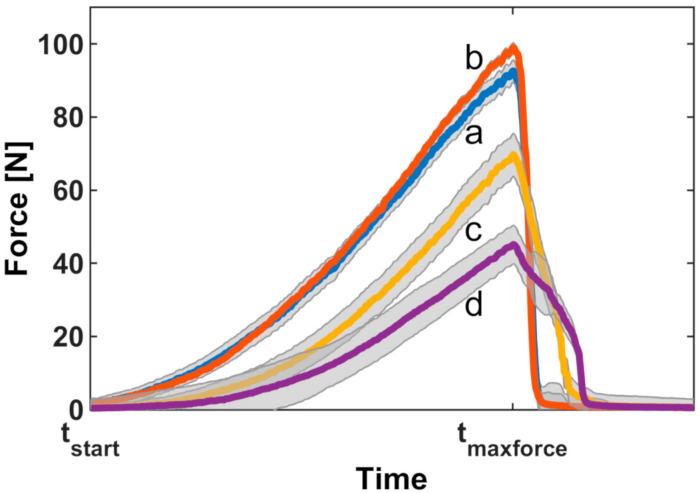
Development of dislodgement forces of four securements. Individual curves were aligned to the time points at which the force application started (t_start_) and at which the maximal force was present (t_maxforce_). Shaded areas represent the 95% confidence intervals: a, sterile wound dressing covered by two elastic polyester fleece dressings, one incised; b, sterile wound dressing covered by two elastic polyester fleece dressings, both incised; c, non-bordered transparent polyurethane film dressing; d, bordered polyurethane film dressing (nonwoven polyester with polyurethane film laminate.

**Figure 5 jcm-09-03192-f005:**
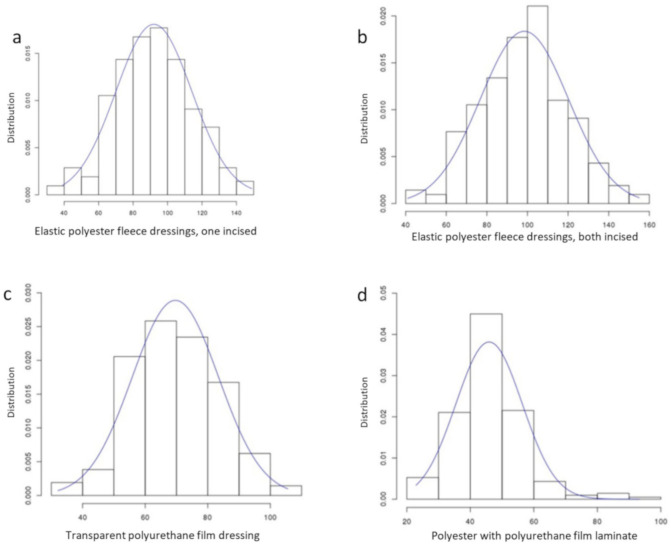
Force distribution of four securement methods: (**a**) elastic polyester fleece dressings, one incised; (**b**) elastic polyester fleece dressings, both incised; (**c**) transparent polyurethane film dressing; (**d**) polyester with polyurethane film laminate. Curves show fitted Gaussian distributions.

**Figure 6 jcm-09-03192-f006:**
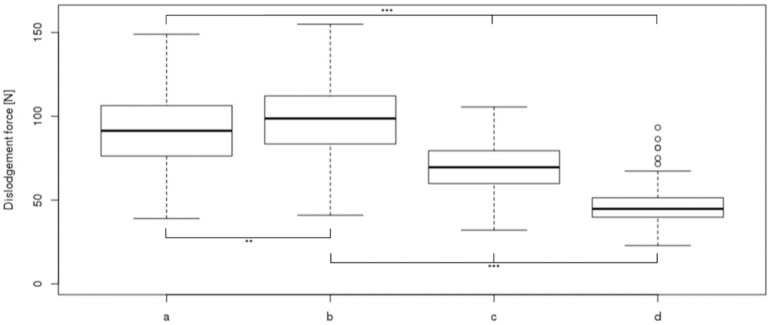
Dislodgement forces of four securements: a, sterile wound dressing covered by two elastic polyester fleece dressings, one incised; b, sterile wound dressing covered by two elastic polyester fleece dressings, both incised; c, non-bordered transparent polyurethane film dressing; d, bordered polyurethane film dressing (nonwoven polyester with polyurethane film laminate). Box plots indicate medians, interquartile range 25–75 and range without outliers (whiskers). Circles show outliers. ** = *p* < 0.01, *** = *p* < 0.001.

**Table 1 jcm-09-03192-t001:** Subjects’ characteristics.

Characteristic	n = 209
Age (years), median (range)	36 (18–73)
Sex (female/male/diverse)	89/120/0

**Table 2 jcm-09-03192-t002:** Costs of four securements.

	Method	Product, Manufacturer	Costs in Euros, Per Dressing
a	Sterile wound dressing covered by two elastic polyester fleece dressings, one incised	Leukomed^®^, BSN medical, Hamburg, GERMANY; 7.2 × 5 cm	0.37
b	Sterile wound dressing covered by two elastic polyester fleece dressings, both incised	Leukomed^®^, BSN medical, Hamburg, GERMANY; 7.2 × 5 cm	0.39
c	Non-bordered transparent polyurethane film dressing	3M™ Tegaderm™ I.V., 3 m, St.Paul, MN, USA; 7 × 8.5 cm	0.65
d	Bordered polyurethane film dressing (nonwoven polyester with polyurethane film laminate)	Curafix^®^ i.v. control, L&R, Frankfurt, GERMANY; 9 × 6 cm	0.44

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
