# Peer review of "Dislodgement Forces and Cost Effectiveness of Dressings and Securement for Peripheral Intravenous Catheters: A Randomized Controlled Trial"

_jcm, 2020, doi:10.3390/jcm9103192_

Round 1

Reviewer 1 Report

Dear author and editor.

Thank you for the opportunity to review your manuscript.

The trial investigate the efficacy of different PIVC dressings applied to healthy test individulas. The choice of dressing was randomised between test persons. No cross-over design was used.

General remarks:

The subject is relevant and for everyone Active in Clinical medicine, the importance of appropriate dressing is obvious.

The mansucript is overall good, but there are some flaws that must be adressed.

Specific remarks:

The use of abbreviation at the beginning of a sentence - please check with the editors if this appropriate.

LL48: Please rephrase, the economic burden appears over-exaggerated.

p 4 2.6: How was the force sensor attached?

LL134: Please provide a rationale as to why staff-cost was excluded. I would assume that a dressing composed of several parts would take longer to attach, thus be more expensive due to staff-consumtion.

LL146: Please present the distribution of test subjects across randomisation Groups. Can you provide more test subject characteristics (other than age and gender?)

LL150: Time (arb,) - meaning??

LL 174: You present gender differences, but what about ttest subject AGE? did that have an impact on the results?

LL202: Lack of staff cost must be added to limitations, as well as only one person applied the dressings (this individual would not represent your average Clinical user.

LL214: Staff is expensive, it is inappropriate to claim cost effectiveness without including this.

Author Response

We express our appreciation to the Reviewer for the time and effort spent in reviewing our work critically. We very much hope that the following replies and the revision of the manuscript will appropriately address the Reviewer´s concerns.

Please see attachment for our response to the comments of the Reviewer.

KInd regards,

Axel Schmutz

Reviewer 2 Report

I commend the authors on this interesting and important work. I have a few questions to be considered in the discussion, including suggestions for future research.

Line 45 & 46: The comment from O'Grady regarding 'a systematic evaluation studying the prevention of catheter dislodgement is still missing' was published in 2011. Since then, several trials of catheter securement have been published, so this sentence should be modified or removed. The issue is still unresolved, as noted in the following sentence.

I'm interested to know if the effect of pulling forces on skin integrity or age was studied. I understand the mean age of the healthy volunteers was 36. Older people have poorer skin elasticity and could suffer more skin tears by skin stretching and pulling forces, which could affect the results. Therefore the authors could recommend that trials on fragile skin are also needed.

What was the time difference between applying the different types of dressings? The Fixomull securement would obviously take more time, so this should be noted.

The CDC 2011 guidelines recommend changing sterile gauze and tape dressings every 2 days, compared with transparent dressings every 7 days. This has cost implications for your study, making the costs of the sterile gauze and Fixomull potentially MORE expensive than the commercial dressings if the IV stays in for a week. In surgical patients, most will not need their IV longer than 48 hours, so the gauze dressing could be more appropriate in this population.

Nurses like the transparent dressings so they can assess for redness or swelling. Perhaps another dressing and securement option for future trials could be a transparent dressing with additional Fixomull securement. 

Finally, was there any dressing preference expressed by the volunteers? It would be good to get some research on patient experience of different dressings, as well.

I understand the authors may choose not to address these issues, but I believe it will make for a more robust paper if they do so.

Copy editing:

Line 35: (Rickard et al, 2018) is not in line with the numbered formatting of the other references.

Line 41: 'und' should be 'and'

Line 66: Author's name should not appear here. Trial registration number is sufficient

Line 85: 3m should be 3M

Lines 153, 162, 169: remove space between 'incised' and ';'

Line 186: 'secure venous intravascular catheter' should be 'secure i.v. catheter'

Dislodgment and dislodgement are both used throughout the paper. Please choose one or the other for consistency.

I enjoyed reading this paper and once again thank the authors for such an interesting and well written paper.

Author Response

We express our appreciation to the Reviewer for the time and effort spent in reviewing our work critically. We very much hope that the following replies and the revision of the manuscript will appropriately address the Reviewer´s concerns.

Kind regards,

Axel Schmutz

Round 2

Reviewer 1 Report

Hi and thanks for letting me review your revised manuscript. I am happy to see that the authors have clarified the adressed issues.